# Incidence, Clinical Presentation and Trends in Indication for Diagnostic Work-Up of Small Intestinal and Pancreatic Neuroendocrine Tumors

**DOI:** 10.3390/diagnostics11112030

**Published:** 2021-11-03

**Authors:** Anna Bryan Stensbøl, Jesper Krogh, Pernille Holmager, Marianne Klose, Peter Oturai, Andreas Kjaer, Carsten Palnæs Hansen, Birgitte Federspiel, Seppo W. Langer, Ulrich Knigge, Mikkel Andreassen

**Affiliations:** 1ENETS Center of Excellence, Rigshospitalet, Copenhagen University Hospital, 2100 Copenhagen, Denmark; anna1bryan@gmail.com (A.B.S.); Jesper.krogh.01@regionh.dk (J.K.); pernille.holmager.01@regionh.dk (P.H.); Marianne.christina.klose.01@regionh.dk (M.K.); Peter.Sandor.Oturai@regionh.dk (P.O.); andreas.kjaer@regionh.dk (A.K.); Carsten.palnaes.hansen@regionh.dk (C.P.H.); birgitte.fe@gmail.com (B.F.); Seppo.Langer@regionh.dk (S.W.L.); ulrich.peter.knigge@regionh.dk (U.K.); 2Department of Endocrinology and Metabolism 7562, Rigshospitalet, Copenhagen University Hospital, 2100 Copenhagen, Denmark; 3Department of Clinical Medicine, University of Copenhagen, 2100 Copenhagen, Denmark; 4Department of Clinical Physiology, Nuclear Medicine & PET and Cluster for Molecular Imaging, Rigshospitalet, Copenhagen University Hospital, 2100 Copenhagen, Denmark; 5Department of Surgery and Transplantation, Rigshospitalet, Copenhagen University Hospital, 2100 Copenhagen, Denmark; 6Department of Pathology, Rigshospitalet, Copenhagen University Hospital, 2100 Copenhagen, Denmark; 7Department of Oncology, Rigshospitalet, Copenhagen University Hospital, 2100 Copenhagen, Denmark

**Keywords:** small intestinal and pancreatic neuroendocrine tumors, incidence, clinical presentation, incidentaloma

## Abstract

Background: The incidence of small intestinal (SI) and pancreatic neuroendocrine tumors (siNETs and pNETs) seems to have increased. The increased frequency of incidental findings might be a possible explanation. The study aimed to examine (1) changes in incidence and the stage at diagnosis (2010–2011 vs. 2019–2020), (2) changes in the initial indication for diagnostic workup and 3) the differences in stage between incidentally discovered vs. symptomatic disease during the entire study period. Methods: We performed a retrospective study, that includes consecutive siNET and pNET patients referred to the Copenhagen ENETS center of excellence in 2010–2011 and 2019–2020. Results: The annual incidence of siNET per 100,000 increased from 1.39 to 1.84, (*p* = 0.05). There was no change in the stage at diagnosis, and in both periods approximately 30% of patients were incidentally diagnosed (*p* = 0.62). Dissemination was found in 72/121 (60%) of symptomatic vs. 22/50 (44%) of incidentally discovered SI tumors in the entire cohort, (*p* = 0.06). The annual incidence of pNET increased from 0.42 to 1.39 per 100,000, (*p* < 0.001). The proportion of patients with disseminated disease decreased from 8/21 (38%) to 12/75 (16%), (*p* = 0.02) and the number of incidental findings increased from 4/21 (19%) to 43/75 (57%), (*p* = 0.002). More symptomatic patients had disseminated disease compared to patients with incidentally discovered tumors (15/49 (31%) vs. 5/47 (11%), (*p* = 0.01)). Conclusion: The incidence of siNET and pNETs increased over the past decade. For siNETs, the stage of disease and the distribution of symptomatic vs. incidentally discovered tumors were unchanged between the two periods. Patients with pNETs presented with more local and incidentally discovered tumors in the latter period. Patients with incidentally discovered siNETs had disseminated disease in 44% of the overall cases. The vast majority of incidentally found pNETs were localized.

## 1. Introduction

Gastroenteropancreatic (GEP) neuroendocrine neoplasms (NENs) represent a heterogeneous group of neoplasms. The diagnosis is primarily based on immunohistochemical staining for the neuroendocrine tumor markers, synaptophysin and chromogranin A (CgA). Another characteristic of well-differentiated neuroendocrine tumor cells is the expression of the somatostatin receptor subtype 2, suggesting that somatostatin receptor imaging (SRI) may be useful in the diagnostic phase [1,2,3].

The World Health Organization (WHO) divides GEP-NENs into well-differentiated neuroendocrine tumors (NETs), and poorly differentiated neuroendocrine carcinomas (NECs). Furthermore, NETs are divided into low-grade (G1, Ki-67 < 3%), medium-grade (G2, Ki-67 3–20%) and high-grade (G3, Ki-67 > 20%) tumors, according to the Ki-67 proliferation index, calculated from the nuclear protein Ki-67 [4,5]. GEP-NENs primarily originate from the pancreas and mid-gut, but can occur in all parts of the gastrointestinal (GI) tract. Functioning NENs (typically G1 and G2) produce and release hormones and amines into the bloodstream, causing specific endocrine symptoms [6,7], whereas non-functioning NENs are hormonally inactive and thus do not cause specific syndromes.

According to the Surveillance, Epidemiology, and End Results (SEER) Program Database, the incidence of NETs has significantly increased over the past 40 years [8]. The increase exceeds the general increase in the incidence of malignant neoplasms [8]. The most recent data has demonstrated an age-adjusted annual increase in incidence of siNETs from 0.20 (1973) to 1.25 (2012) per 100,000 inhabitants, whereas the incidence of pNETs increased from 0.18 (1973) to 0.81 (2012) [8]. Results from Europe and Canada have revealed similar tendencies [9,10]. Incidences of si- and pNETs have not been reported after 2012. An increased use of imaging, detecting more incidental tumors, has been suggested as the primary cause for the observed increase in incidence. A greater awareness among pathologists as well as a genuine increase in incidence have also been suggested as potential causes [8,9,10,11,12]. However, these suggestions are mostly based on single case studies and are not supported by scientific evidence. In several studies, it has been shown that the vast majority of small (<2 cm) incidentally discovered pNETs (incidentalomas) show a benign growth pattern, suggesting watchful waiting as a feasible follow-up strategy [13,14,15,16]. To the best of our knowledge, there are no published data exists that documents the number of incidentally discovered siNETs or the stage at diagnosis in incidentally discovered tumors.

In the present retrospective study, we set out to investigate G1 and G2 si- and pNET patients diagnosed between 2010–2011 and 2019–2020. The primary endpoints were (1) changes in overall incidence and stage at diagnosis (2010–2011 vs. 2019–2020), (2) changes (2010–2011 vs. 2019–2020) in the initial indication for diagnostic workup, focusing on symptomatic disease vs. incidental findings and (3) differences in stage at diagnosis between incidentally discovered vs. symptomatic disease during the entire study period (2010, 2011, 2019, and 2020).

## 2. Methods

### 2.1. Study Population

Patients (≥18 years) with a diagnosis of siNET or pNET, who were referred to the NET Center (European Neuroendocrine Tumor Society (ENETS) Center of Excellence), Rigshospitalet, Copenhagen University Hospital during 2010–2011 and 2019–2020 were included. The Center receives all patients from Eastern Denmark (Capital Region of Denmark and Zealand Region), covering approximately half of the total Danish population. During the period of 2010–2020, the population in Eastern Denmark increased by approximately 175,000 inhabitants (2010: 2,517,798 and 2020: 2,692,425). The study was based on data collected from patient files.

Inclusion criteria were as follows: First-time histological or radiological verified G1 or G2 siNET or pNET. The histological diagnosis was based on immunohistochemical staining for synaptophysin and chromogranin-A (CgA) on surgically resected specimens or core biopsies. If a patient declined biopsy or it was not feasible to perform one, a diagnosis was established if a tumor was visible on either CT or MRI *and* SRI (with high tracer uptake, Krenning grade 3–4 [17,18]). The exclusion criterion included mixed neuroendocrine non-neuroendocrine neoplasms (MiNEN).

At the NET center, all patients were discussed at a multidisciplinary tumor board with the participation of a pathologist, a radiologist, a nuclear physician, an endocrinologist, an oncologist and a surgeon, all of whom specialized in NET tumors. In addition, all the histological material was re-evaluated by a dedicated NET pathologist at Rigshospitalet, Copenhagen. In the case of endocrine symptoms, relevant gastrointestinal hormones were measured. Tumor tissue was stained for Ki-67 with monoclonal mouse anti-human Ki-67 antigen clone MIB-1 (Dako^®^, Agilent, Santa Clara, CA, USA), and the mean percentage of Ki-67-labeled nuclei was measured in 20 hotspot regions [19]. Measurements of the p-CgA were performed by an in-house radioimmunoassay (CgA340-348) and were obtained before surgery [20].

### 2.2. Definitions

Small intestinal NETs were defined as NETs originating from the jejenum, ileum and cecum. If histological material was present, the tumor tissue would stain for serotonin. Pancreatic NETs were defined as NETs originating from the pancreas and duodenum.

Incidentally discovered NETs were defined as NETs found by a scan or endoscopy requested on the basis of a non-neuroendocrine disease or the discovery of a NET in a surgical specimen operated on for another disease. Symptomatic NETs were defined as NETs diagnosed due to relevant symptoms, NETs presenting with acute symptoms requiring surgery or NETs detected during surveillance for Multiple Endocrine Neoplasia 1 (MEN1). The relevant symptoms leading to the initial investigation were categorized into the following groups: Gastrointestinal symptoms (diarrhea, constipation, abdominal pain, dyspepsia, nausea, vomiting or for siNETs; bloody stools), flushing, attack-like phenomena (palpitations, hypertension, dyspnea, symptoms of hypoglycemia or excessive sweating) and unspecific symptoms (night sweat, weight loss, fever, tiredness/fatigue, jaundice, anemia, lymphadenopathy or pain related to potential metastases).

The stage of disease at diagnosis was divided into the following categories: Localized disease (no disease spread), regional disease (growth into a neighboring structure or in lymph nodes below diaphragm) and disseminated disease (metastasis to liver or any other organ, bone metastasis, carcinomatosis or lymph nodes above diaphragm). The plasma CgA measurement was excluded from the analysis if the sample was obtained after surgery had been performed.

### 2.3. Statistics

The data collection and statistical analyses were performed in SPSS (IBM SPSS statistics 27). Descriptive statistics were used to summarize the subject characteristics. Non-normally distributed data are presented as median and interquartile range (IQR). Normally distributed data are presented as mean ± standard deviation (SD). An assessment of the distribution was performed through a visual inspection of histograms. To approximate the normal distribution tumor size, the Ki-67 index and pCgA, the natural logarithm was transformed when calculating *p*-values. The continuous variables were compared by unpaired *T*-test whereas categorical data were compared by χ^2^-tests or the Fisher’s exact test where appropriate.

Incidences in 2010, 2011, 2019 and 2020 were calculated individually as the number of cases divided by the number of inhabitants in the respective year and expressed as cases per 100,000 inhabitants. A mean of the incidence of cases in 2010–2011 and 2019–2020 was calculated and used for the analyses. The *p*-value describing the differences in incidence (2010–2011 vs. 2019–2020) was calculated as a relative-risk in SPSS. The age-adjusted incidence rates were calculated using weighted proportions of the 2010–2011 age groups in both periods (2010–2011 and 2019–2020).

### 2.4. Ethics

The study was approved by the local data protection agency at Rigshospitalet (2007-58-0015) (4 June 2011) and by the Danish Patient Safety Authority (31-1521-453) (14 August 2020). Due to the retrospective design, informed consent was not required.

The data that support the findings of this study are available upon request from the corresponding author. The data are not publicly available due to privacy or ethical restrictions.

## 3. Results

### 3.1. Baseline Characteristics

A total of 875 patients with proven or suspected G1 or G2 NET were referred in two calendar year periods (246 patients in 2010–2011 vs. 629 in 2019–2020). For siNETs, 70 patients fulfilled the inclusion criteria in 2010–2011 vs. 101 in 2019–2020. For pNETs, data from 21 patients were included in 2010–2011 vs. 75 in 2019–2020. There were 6 duodenal tumors among the pNETs (3 in 2010–2011 vs. 3 in 2019–2020). Patient selection is illustrated in Figure 1.

### 3.2. Small Intestinal NETs

Patient characteristics are shown in Table 1. There was no difference in the mean age or gender distribution between the two time periods. The crude average annual incidence rate of siNETs increased from 1.39 in 2010–2011 to 1.84 in 2019–2020 per 100,000, (*p* = 0.05). The annual age-adjusted incidence of siNETs was 1.39 per 100,000 in 2010–2011 vs. 1.76 per 100,000 in 2019–2020, (*p* = 0.30).

Except for four cases in 2019–2020, all patients in both the calendar periods received their diagnosis based on histology. Of the four cases, two patients died before the biopsy, one declined to have a biopsy and in one case it was not possible to visualize the tumor using ultrasound. In 2010–2011 all SRI were ^111^in-octreotide scintigraphy whereas all SRI in 2019–2020 were ^8^Ga-DOTATOC or ^64^Cu-DOTATETE PET/CT (Table 1) [21].

There was no difference in the disease stage at diagnosis in the two calendar periods (Figure 2A and Table 1). A localized disease was found in 2/70 (3%) vs. 7/101 (7%) (*p* = 0.24), a regional disease in 25/70 (36%) vs. 42/101 (42%) (*p* = 0.44) and disseminated disease in 42/70 (60%) vs. 52/101 (51%) (*p* = 0.27). The annual incidence rate of disseminated siNET was 0.80 per 100,000 in 2010–2011 vs. 0.97 in 2019–2020 (*p* = 0.61). No significant differences were found for the two periods when metastases above the diaphragm, bone metastases, tumor size or the Ki-67 index, were compared. Preoperative plasma CgA was 1485 (227–10,575) in 2010–2011 vs. 317 (140–1760) pmol/L in 2019–2020, (*p* = 0.01).

#### Symptomatic Disease vs. Incidental Findings

In 2010–2011 vs. 2019–2020 there was no difference in the occurrence of incidental findings (19/70 (27%) vs. 31/101 (31%)), (*p* = 0.62). In 2019–2020, 6 out of 31 incidentally discovered siNETs were found during the national screening program for colorectal cancer, initiated on 1st of March 2014 (Table 1). The crude annual incidence per 100,000 of the symptomatic siNETs was 1.01 in 2010–2011 vs. 1.30 in 2019–2020 (*p* = 0.37) and the annual incidence of SI incidentalomas was 0.38 in 2010–2011 vs. 0.58 in 2019–2020 (*p* = 0.31).

The disease stage for symptomatic patients (n = 121) vs. for an incidentally diagnosed disease (n = 50) from both periods are presented in Figure 3A. The incidental group had a significantly higher level of regional staged disease compared to the symptomatic group (26/50 (52%) vs. 41/121 (34%), *p* = 0.03), whereas the symptomatic group showed a tendency towards the more disseminated stage of disease (72/121 (60%) vs. 22/50 (44%), *p* = 0.06). There were no significant differences found between the groups when metastases above the diaphragm, bone metastases or tumor size (data not shown) were compared.

PET imaging of an incidentally discovered siNET patient is illustrated in Figure 4.

### 3.3. Pancreatic NETs

Among the patients with pNETs, there were no differences in age (*p* = 0.50) or gender distribution (*p* = 0.23) (Table 2). The crude incidence increased from 0.42 in 2010–2011 to 1.39 per 100,000 in 2019–2020, (*p* < 0.001). The annual age-adjusted incidence of pNETs was 0.42 in 2010–2011 vs. 1.34 per 100,000 in 2019–2020, (*p* < 0.001).

In 2010–2011 all patients, except one, were diagnosed based on histology whereas only 47/75 (63%) were diagnosed based on histology in 2019–2020, (*p* = 0.004). The reason for the lack of a histological diagnose was either that patients refused to have a biopsy done, that it was not feasible to perform a biopsy, or it was not considered clinically relevant due to factors such as age or concomitant diseases. A greater number of the pNETs were functional in 2010–2011 vs. 2019–2020 (7/21 (33%) vs. 10/75 (13%), *p* = 0.03) (Table 2).

There was a major difference in disease stage at diagnosis between the two calendar periods, as 8/21 (38%) had disseminated disease in 2010–2011 vs. 12/75 (16%) in 2019–2020, (*p* = 0.02) (Figure 2B and Table 2). The incidence of dissemination was 0.16 per 100,000 in 2010–2011 vs. 0.22 in 2019–2020 (*p* = 0.60), whereas the incidence of non-disseminated pNETs significantly increased from 0.26 to 1.17, (*p* < 0.001). The tumor sizes were larger and the p-CgA was higher in 2010–2011 vs. 2019–2020, (*p* = 0.03 and *p* = 0.02), (Table 2).

#### Symptomatic Disease vs. Incidental Findings

The proportion of patients with incidental tumors increased from 4/21 (19%) to 43/75 (57%), (*p* = 0.002). Conversely, the proportion of patients initially investigated due to symptoms or MEN1 surveillance decreased from 17/21 (81%) in 2010–2011 to 32/75 (43%) in 2019–2020, (*p* = 0.002). The incidence rate of symptomatic pNETs did not increase (0.34 in 2010–2011 vs. 0.60 in 2019–2020, *p* = 0.22), whereas the incidence rate of the incidentalomas increased from 0.08 (2010–2011) to 0.80 (2019–2020), (*p* = 0.002). One incidentaloma, diagnosed in 2019–2020, was an insulinoma. The other incidentalomas were non-functioning.

The disease stage of symptomatic patients (n = 49) vs. incidentally diagnosed patients (n = 47) from both periods are presented in Figure 3B. A greater number of symptomatic patients had disseminated disease compared to patients with pNET incidentalomas, at 15/49 (31%) vs. 5/47 (11%), (*p* = 0.01). Moreover, the symptomatic tumors were larger than the non-symptomatic tumors (2.0 (1.2–3.8) vs. 1.6 (1.0–2.4) cm, *p* = 0.02).

### 3.4. CT and MR Scans in Denmark

According to data from the Danish National Board of Health, the number of abdominal CT scans in Denmark increased from 48/1000 persons in 2010 to 84/1000 persons in 2020. Likewise, the number of abdominal MRI scans increased from 6/1000 persons in 2010 to 9/1000 persons in 2020.

## 4. Discussion

The three major findings of this retrospective population-based study were (1) the increase in the incidence of siNET and pNET that was observed in previous studies seems to have continued (2) during the last decade, the presentation of pNETs has changed substantially, with more localized tumors incidentally discovered in 2019–2020 compared to the number discovered in 2010–2011. By contrast, stage at diagnosis and clinical presentation for siNET remained unchanged between 2010–2011 and 2019–2020 and (3) a high proportion—almost 50%—of incidentally discovered asymptomatic siNET were disseminated at diagnosis.

The crude siNET incidence increased by 30%, from 1.39 to 1.84 per 100,000, whereas the crude pNET incidence increased by 350%, from 0.42 to 1.39, in the years 2010–2020. In comparison, an American (year 2012) [8], a Norwegian (year 2010) [9] and a Canadian (year 2009) [10] study presented an age adjusted siNET incidence of 1.25, 1.31 and 1.01 and a pancreatic age adjusted incidence of 0.81, 0.71 and 0.69. Because of the different methods for age adjusting, the presented incidences cannot be directly compared to the previous data. However, the results strongly suggest a continuous increase in si- and pNET incidences from 2010 to 2020.

For siNETs, the stage at diagnosis did not change from 2010–2011 to 2019–2020. In both periods, approximately half of the patients had disseminated disease at diagnosis. Thus, in terms of earlier detection, it seems that there has been no improvement in siNET diagnostics in recent decades. In agreement with the current data, a Norwegian study found that the mean percentage of patients with disseminated disease, in the period 1993–2010, was 51% [9]. Plasma-CgA levels in siNET patients were significantly higher in 2010–2011 compared to 2019–2020, suggesting a higher disease burden in the former period. However, in the latter period, patients were advised to pause proton pump inhibitors before taking measurements of p-CgA, which probably contributes to the lower Plasma-CgA levels observed in 2019–2020.

For pNETs, substantial differences in the stage at diagnosis were observed between 2010–2011 vs. 2019–2020. In the former period, a significantly higher fraction of patients had disseminated disease, the primary tumors were larger and the p-CgA levels were higher. A decrease in the metastatic appearance and the smaller tumor sizes may suggest that pNETs are now detected earlier than they were 10 years ago, presenting the potential of better prognosis. However, numerically, there were no significant differences in the number of patients with disseminated disease between the two periods (8 vs. 12). Given the indolent nature of these tumors, the detection of small pNETs and their subsequent inclusion in imaging surveillance programs may only have a minimal impact on the prognosis for the individual patient.

Several studies have investigated the rising incidence of both siNET and pNET and different explanations have been proposed [8,9,10]. To the best of our knowledge, this is the first study to systematically examine the change in the proportion of si- and pNET patients diagnosed due to symptoms vs. incidentally discovered disease over time. The increase in incidence of siNETs does not seem to be explained by incidental findings, since the proportion of patients incidentally diagnosed in 2010–2011 vs. 2019–2020 was unchanged. Thus, there may have been a genuine increase in siNET incidence over the last 10 years, with an aging population as a contributing factor. In the cohort, nearly 30% of the siNETs in both periods were incidental findings. This is, to our knowledge, a novel observation. In 2019–2020, 6 of 101 patients were diagnosed because of the newly introduced surveillance program in Denmark, which used blood in stools as the initial screening method and a subsequent colonoscopy in the case of a positive test.

In contrast to siNETs, the data showed a highly significant increase in pancreatic incidentalomas from 19% in 2010–2011 to 57% in 2019–2020—in nominal numbers from 4 to 43. In comparison, a Norwegian study found that 19% of pNETs diagnosed in 1982–2010 were incidentalomas [22] and an Italian study found 39% to be incidentally diagnosed in 2004–07 [23]. The number of CT and MRI scans almost doubled from 2010 to 2020, partially explaining the increase in pancreatic incidentalomas. More awareness among radiologist and improved CT technology might be another contributing factor. ^68^Ga-DOTATOC/^64^Cu-DOTATATE-PET/CT was exclusively performed post-referral at the NET Center and meaning that the shift from SPECT to the more sensitive PET did not contribute to the increase in incidentally discovered pNETs [24].

As the presented data illustrate, incidental findings on imaging constitute an increasing challenge in everyday clinical practice. It has been shown that neuroendocrine incidentalomas originating from areas such as the pituitary gland, adrenal medulla or appendix seem to have a more indolent behavior compared to the symptomatic ones of the same primary site [8,25,26]. In contrast to these findings, our data show that incidental discovered siNETs seem to have a phenotype comparable to the symptomatic discovered disease. Forty-four percent of patients with incidentally discovered disease were disseminated at the time of diagnosis vs. 60% of the symptomatic patients. Generally, only disseminated siNETs with liver metastases give rise to hormonal symptoms and, as a result of everyday clinical practice, it is well known that the general symptoms of a malignant disease such as pain, weight-loss and nausea are often absent in patients with disseminated siNETs, explaining the high number of metastases among incidentally discovered siNETs.

In contrast to the results from siNETs, patients with symptomatic pNETs were three times more likely to have disseminated disease compared to patients with pancreatic incidentalomas, and they presented with larger tumors. Likewise, a Norwegian study found that pNET incidentalomas were significantly smaller compared to those that were symptomatic [22], and an Italian study found that the presence of liver metastases was significantly higher in symptomatic compared to asymptomatic patients [23]. Therefore, ENETS’ guidelines for non-functioning pancreatic G1 and low G2 NET recommend that pNETs < 2 cm should be managed through active surveillance instead of surgery [13,14,15,16]. ENETS’ guidelines are followed at Rigshospitalet, Copenhagen, explaining why many incidentalomas were not resected. Further research must focus on the optimal surveillance of pNET incidentalomas to avoid causing a needless burden on patients (e.g., imaging with radiation exposure and anxiety for the patients).

The main strengths of this study include the patient file-based design, making it possible to investigate specific symptoms, the diagnostic procedures, and the detailed patient characteristics prior to referral. This makes the study independent of registers and databases that can be erroneous and incomplete. Furthermore, it ensures that all data are recorded in the same way and thus reduces the risk of bias. The limitations are that the data were collected retrospectively, which can result in incomplete or unavailable data as well as a change in clinical awareness and protocols. Another limitation is the size of the study as it included a relatively small number of patients, implying the risk of type 2 statistical errors. Only patients referred to the NET center were included in the study, but it must be acknowledged that a few patients diagnosed with si- or pNET refused referral or were not candidates for referral due to a poor performance status. Finally, it must also be acknowledges that the results are somewhat biased due to a change in the imaging methodologies between the two periods. This was most apparent in the shift from traditional somatostatin receptor scintigraphy to PET technology with ^68^Ga-DOTATOC/^64^Cu-DOTATE-PET, which may have improved the sensitivity and detection rate, implying that more patients were diagnosed with disseminated disease in 2019–2020 [13,24,27].

In conclusion, for siNETs, there was an increase in incidence, whereas the stage at diagnosis and the clinical presentation remained unchanged between 2010–2011 and 2019–2020. Furthermore, the incidence of pNETs more than tripled, primarily driven by a large increase in localized incidentalomas. Almost half of the incidentally discovered siNETs were disseminated, highlighting the need for a thorough diagnostic work-up of these patients. The vast majority of the incidentally discovered pNETs were small and localized and future studies should focus on the optimal method for follow-up of these patients.

## Figures and Tables

**Figure 1 diagnostics-11-02030-f001:**
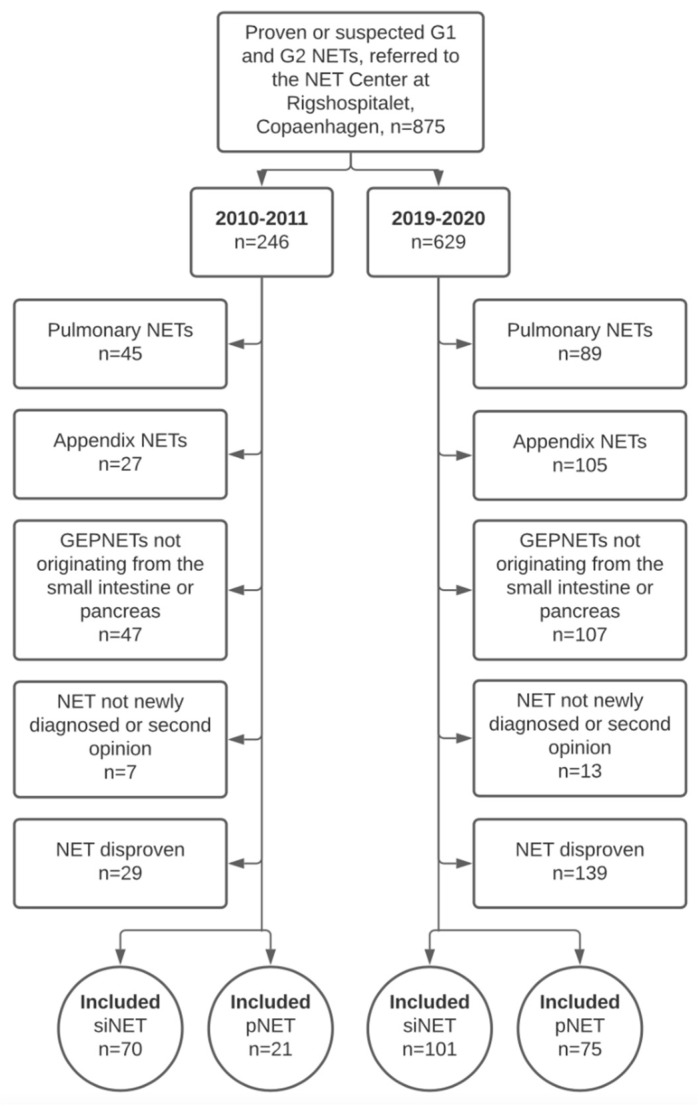
Flowchart illustrating patient selection. Flowchart illustrating excluded patients and the number of small intestinal and pancreatic NETs included. Abbreviations: Neuroendocrine Tumor (NET). Small intestinal NET (siNET). Pancreatic NET (pNET) Gastroenteropancreatic NET (GEPNET).

**Figure 2 diagnostics-11-02030-f002:**
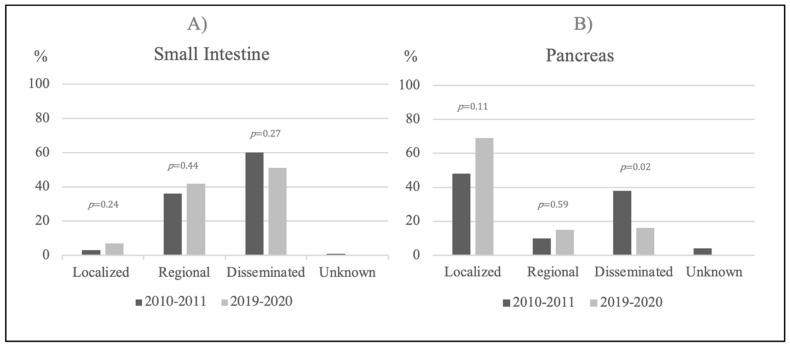
Time trends in disease stage at referral. (**A**) Time trends in disease stage at referral of small intestinal NETs in the two calendar years 2010–2011 and 2019–2020. (**B**) Time trends in disease stage at referral of pancreatic NETs in the two calendar years 2010–2011 and 2019–2020. The *p*-value illustrates the difference between the calendar periods. Abbreviation: Neuroendocrine Tumor (NET).

**Figure 3 diagnostics-11-02030-f003:**
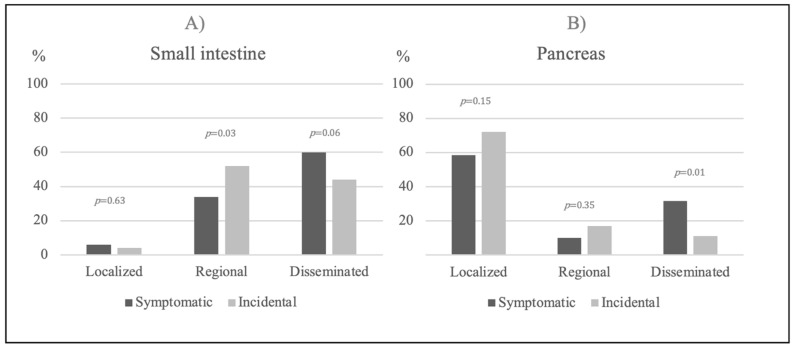
Disease stage symptomatic vs. incidental findings in the entire cohort. (**A**) Disease stage of small intestinal NETs when comparing symptomatic vs. incidental findings in the entire cohort (2010, 2011, 2019 and 2020). (**B**) Disease stage of pancreatic NETs when comparing symptomatic vs. incidental findings in the entire cohort. The *p*-value illustrates the difference between the symptomatic and incidental findings. Abbreviation: Neuroendocrine Tumor (NET).

**Figure 4 diagnostics-11-02030-f004:**
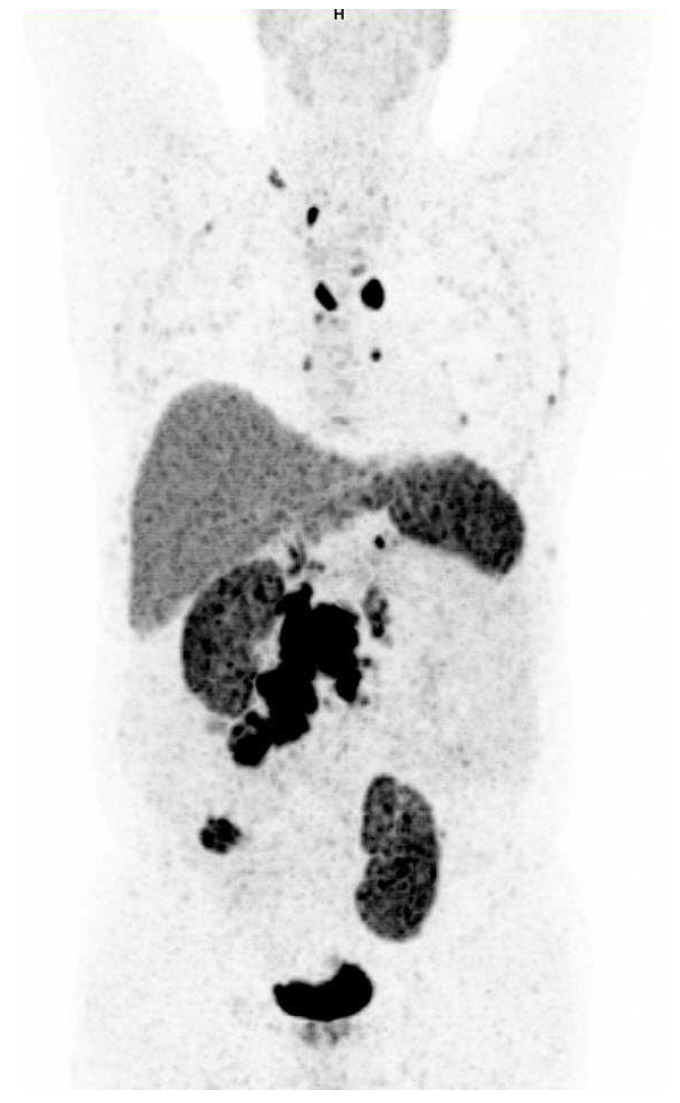
Imaging of a patient with incidentally discovered small intestinal NET. ^68^Ga-DOTATOC-PET from a patient with disseminated incidentally discovered small intestinal NET in a seventy-five-year-old man. As part of the investigation for prostate cancer, an MRI of the prostate was performed, and a tumor in the abdomen was incidentally discovered. In retrospect he reported loose stools for a year. He did not have flushing or other general symptoms of malignant disease. The patient had a primary tumor of 2.9 cm placed in right lower quadrant (illustrated by the arrow), a 5.5 cm mesentery metastasis and multiple lymph node metastasis. Moreover, the patient presented with metastasis in the sternum, left scapula, left ileal bone and in mediastinal lymph nodes. The picture has been published with the patient’s permission.

**Table 1 diagnostics-11-02030-t001:** Baseline characteristics of small intestinal NET patients.

**Small Intestinal NET, Pre Referral**			
	**2010–2011**	**2019–2020**	***p*-Value**
**Number (n) of Cases**	70	101	
**Age**	65 ± 12	65 ± 12	0.99
**Female, n (%)**	36 (51)	44 (44)	0.31
**Incidence (per 100,000)**	1.39	1.84	0.05
**Age adjusted incidence (pr 100,000)**	1.39	1.76	0.30
**Incidental findings, n (%)**	19 (27)	31 (31)	0.62
Incidental finding on imaging or endoscopy, n (%)	12 (17)	24 (24)	0.30
Incidental finding after surgery, n (%)	7 (10)	7 (7)	0.47
**Symptoms leading to initial investigation**	51 (73)	70 (70)	0.62
GI symptoms, n (%)	28 (40)	43 (43)	0.74
GI symptoms + flushing, n (%)	5 (7)	7 (7)	0.96
Flushing, n (%)	2 (3)	2 (2)	0.71
Unspecific symptoms, n (%)	6 (9)	5 (5)	0.34
Acute surgery for ileus, n (%)	10 (14)	13 (13)	0.79
**Histologic diagnosis at time of referral to NET center (%)**	62 (89)	81 (80)	0.15
**Small Intestinal NET, Post Referral**			
**Diagnosis based on**			
Histology, n (%)	70 (100)	97 (96)	0.09
Imaging, n (%)	0	4 (4)	0.09
**^68^Ga-DOTATOC PET-CT/^64^Cu-DOTATATE PET-CT, n (%)**	0	99 (98)	<0.001
**^111^In-octreotide scintigraphy, n (%)**	65 (93)	0 (0)	<0.001
**Stage**			0.30
Localized, n (%)	2 (3)	7 (7)	0.24
Regional, n (%)	25 (36)	42 (42)	0.44
Disseminated, n (%)	42 (60)	52 (51)	0.27
Unknown, n (%)	1 (1)	0	-
**Metastases above diaphragm, n (%)**	10 (14)	22 (22)	0.22
**Bone metastases, n (%)**	4 (6)	11 (11)	0.24
**Tumor size (cm)**	2.5 (1.5–3.5)	1.9 (1.4–2.9)	0.06
**p-CgA (before surgery) nm/L**	1485 (227–10,575)	317 (140–1760)	0.01
**Ki-67 index (tumor or metastasis), (%)**	4 (2–6)	4 (2–7)	0.13

Patient and tumor characteristics by year of diagnosis. Tumor size is based on 45/70 (2010–2011) and 71/101 (2019–2020) patients. p-CgA is based on 40/70 (2010–2011) and 67/101 (2019–2020). Ki-67 index is based on 69/70 (2010–2011) and 97/101 (2019–2020). Among the incidental findings, the most common indication for imaging was surveillance for other types of cancer. Abbreviations: Neuroendocrine Tumor (NET). Gastro-intestinal (GI). Plasma-Chromogranin A (p-CgA).

**Table 2 diagnostics-11-02030-t002:** Baseline characteristics of pancreatic NET patients.

**Pancreatic NET, Pre Referral**			
	**2010–2011**	**2019–2020**	** *p* ** **-Value**
**Number (n) of Cases**	21	75	
**Age**	64 ± 15	61 ± 13	0.50
**Female/male**	7/14	36/39	0.23
**Incidence (pr 100,000)**	0.42	1.39	<0.001
**Age adjusted incidence (pr 100,000)**	0.42	1.34	<0.001
**Incidental findings, n (%)**	4 (19)	43 (57)	0.002
Incidental finding on imaging, n (%)	4 (19)	37 (49)	0.01
Incidental finding after surgery, n (%)	0	6 (8)	0.18
**Symptoms/reason leading to initial investigation**	17	32	0.002
GI-symptoms, n (%)	10 (48)	9 (12)	<0.001
Attack-like phenomena, n (%)	3 (14)	8 (11)	0.66
Unspecific symptoms, n (%)	4 (19)	11 (15)	0.63
MEN1 follow up, n (%)	0	4 (5)	0.28
**Histologic diagnosis at time of referral to NET center (%)**	13 (62)	28 (37)	0.04
**Pancreatic NET, Post Referral**			
**Type of tumor**			
Non-functioning, n (%)	14 (67)	65 (87)	0.03
Functioning, n (%)	7 (33)	10 (13)	0.03
**Diagnosis based on**			
Histology, n (%)	20 (95)	47 (63)	0.004
Imaging, n (%)	1 (5)	28 (37)	0.004
** ^68^ ** **Ga-DOTATOC PET-CT/^64^Cu-DOTATATE PET-CT, n (%)**	0	68 (91)	<0.001
** ^111^ ** **In-octreotide scintigraphy, n (%)**	18 (86)	0 (0)	<0.001
**Stage**			0.03
Localized, n (%)	10 (48)	52 (69)	0.11
Regional, n (%)	2 (10)	11 (15)	0.59
Disseminated, n (%)	8 (38)	12 (16)	0.02
Unknown, n (%)	1 (4)	-	-
**Metastases above diaphragm, n (%)**	3 (14)	6 (8)	0.64
**Bone metastases, n (%)**	0	4 (5)	0.18
**Tumor size (cm)**	2.5 (1.8–8.3)	1.6 (1–2.6)	0.03
**p-CgA (before surgery) nm/L**	259 (83–1590)	92 (68–170)	0.02
**Ki-67 Index (tumor or metastasis), %**	5 (2–8)	4 (2–9)	0.56

Patient and tumor characteristics by year of diagnosis. Tumor size is based on 18/21 (2010–2011) and 66/75 (2019–2020) patients. The p-CgA is based on 15/21 (2010–2011) and 65/75 (2019–2020). The Ki-67 index is based on 19/21 (2010–2011) and 47/75 (2019–2020). Among the incidental findings, the most common indication for imaging was surveillance for other types of cancer. Abbreviations: Neuroendocrine Tumor (NET). Gastro-intestinal (GI). Plasma-Chromogranin A (p-CgA). Multiple endocrine neoplasia type 1 (MEN1).

## Data Availability

The data that support the findings of this study are available on request from the corresponding author. The data are not publicly available due to privacy or ethical restrictions.

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
