# Peer review of "Incidence, Clinical Presentation and Trends in Indication for Diagnostic Work-Up of Small Intestinal and Pancreatic Neuroendocrine Tumors"

_diagnostics, 2021, doi:10.3390/diagnostics11112030_

Round 1

Reviewer 1 Report

the manuscript ought to to investigate G1 and G2 si and p NET patients diagnosed in 2010-11 and 2019-2020 in a retrospective assett. to me, it has a scientific soundness and catches the readers' attention. minor revisions are reuired to improve the manuscript and bring it to publication such as grammar improvements, text editing, spoken languange reviews.

Author Response

Dear reviewer 1

Thank you for the suggestions to improve the manuscript, we have revised the manuscript according to the suggestions

Best regards 

Mikkel Andreassen 

Reviewer 2 Report

REVIEW                                                                              

Manuscript:

Incidence, clinical presentation and trends in indication for diagnostic work-up of small intestinal and pancreatic neuroendocrine tumors.

The authors report on changes in incidence and stage at diagnosis, changes in initial indication for diagnostic work-up and differences in stage between incidental discovered vs. symptomatic small intestinal and pancreatic neuroendocrine tumors. It is an interesting presentation and important topic in this rare disease with increasing incidence.

The study has some limitations due to its retrospective nature, small sample size and changes in imaging between the study periods.

On the other hand, the manuscript is well-prepared and completed by excellent tables and figures.

I would recommend the editors to accept the manuscript in the present form.

Yours Sincerely

Author Response

Dear reviewer 2

Thank you for appriciation  of our results  

Best regards 

Mikkel Andreassen 

Reviewer 3 Report

well conducted study no more comment

congratulations 

Author Response

Dear reviewer 3

Thank you for appreciation  of our results  

Best regards 

Mikkel Andreassen